# Deletion of the *OsLA1* Gene Leads to Multi-Tillering and Lazy Phenotypes in Rice

Zhanglun Sun [1], Tianrun Mei [1] , Tingting Feng [1], Hao Ai [1] , Yafeng Ye [2], Sumei Duan [1], Binmei Liu [2,*] and Xianzhong Huang [1,*]

1   Center for Crop Biotechnology, College of Agriculture, Anhui Science and Technology University, Chuzhou 239000, China; sunzl127425@126.com (Z.S.); mtr0322@foxmail.com (T.M.); fengtingtingyt@ahstu.edu.cn (T.F.); aihao@ahstu.edu.cn (H.A.); duansm@ahstu.edu.cn (S.D.)
2   Key Laboratory of High Magnetic Field and Ion Beam Physical Biology, Hefei Institutes of Physical Science, Chinese Academy of Sciences, Hefei 230001, China; yyfeng@ipp.ac.cn
*   Correspondence: liubm@ipp.ac.cn (B.L.); huangxz@ahstu.edu.cn (X.H.)

**Abstract:** Plant architecture, one of the key factors that determine grain yield in rice, is mainly affected by components such as plant height, tiller number, and panicle morphology. For this paper, we obtained a multi-tillering and lazy mutant from a *japonica* rice cultivar, Wuyunjing 7 (WYJ7), via treatment with a heavy ion beam. Compared to WYJ7, the mutant showed a significant increase in tiller angle, tiller number, number of primary and secondary branches, and number of grains; however, the plant height and grain thickness of the mutant was significantly decreased. Phenotypic analysis of the $F_1$ hybrids revealed that the multi-tillering and lazy mutant phenotypes were regulated by a recessive gene. The segregation ratio of 1:3 of the mutant phenotype and the wild-type plant in the $F_2$ population indicated that the former was controlled by a single gene named *Multi-Tillering and Lazy 1* (*MTL1*). Bulked segregant analysis was performed using the individual plants with extremely typical tiller angles in the $F_2$ population. The *MTL1* gene was initially mapped within a region of 5.58–17.64 Mb on chromosome 11. By using the $F_2$ segregated population for fine mapping, the *MTL1* gene was ultimately fine mapped within the range of 66.67 kb on chromosome 11. The analysis of genes in this region revealed the presence of the previously identified *LAZY1* (*LA1*) gene. Genomic PCR amplification and semi-quantitative RT-PCR assays showed that the *LA1* gene could not be amplified and was not expressed, thus indicating that the *MTL1* gene might be identical to the *LA1* gene. This study suggests that the multi-tillering and lazy mutant phenotypes might be caused by the deletion of *LA1* function. This finding can guide further investigations on the functional mechanisms of the *LA1* gene, thus enriching the theoretical knowledge of plant architecture in relation to rice.

**Keywords:** rice; plant architecture; tillering; LAZY; map-based cloning



## 1. Introduction

As one of the most important food crops in the world, rice (*Oryza sativa* L.) is a staple food for more than half of the world's population. Rice yield is an important factor that can be adapted to meet the increasing global demand for food among the world's population. The plant height, tiller number, tiller angle, and panicle morphology of rice determine the plant's architecture, a crucial factor for yield formation [1]. In the 1960s, based on *Semidwarf 1* (*SD1*) mutation, the identification and utilization of semi-dwarf varieties greatly increased the yield of rice [2–4]. Therefore, achieving the ideal plant structure plays an important role in improving rice yield.

Tiller is one of the important factors that determine rice plant architecture and yield. Tillering mainly includes tiller number, tiller angle, and tiller ability. Previous studies have shown that rice tillers are affected by both environmental and genetic factors, among which genetic factors are crucial [5]. In recent years, several genes that regulate tiller-related traits in rice have been identified. The first gene cloned in rice to control rice tillers was

*MONOCULM 1* (*MOC1*), which is a homologous gene of *LATERAL SUPPRESSOR* (*LAS*) in tomato and *Arabidopsis*; this gene encodes a GRAS family nucleoprotein [6–8]. The mutant *moc1* does not have any tillers; it has only one main stem, and the inflorescence axis and panicles are also smaller than those of the wild-type plant [8]. The overexpression of the *MOC1* gene increased tillers and led to a high level of tillering [8]. The *MOC2* gene encodes the enzyme fructose-1,6-diphosphatase. Although the *moc2* mutant produces tiller buds, the growth of these buds is suppressed or they cannot further develop into tillers; this condition manifests via single tillers, dwarfing, light green leaves, short and narrow leaves, small panicles, and reduced grain number per panicle [9]. The *MOC3* gene encodes a WUSCHEL-related homeobox protein family member that functions as a transcription suppressor. The *moc3* mutant cannot form axillary buds and has no tillers; during the reproductive stage, the mutant produces only one inflorescence, and the development of spikelets is affected, resulting in a reduced number of spikelets and several morphological defects [10,11]. The rice *MOC3/WUS* dysfunction mutant *decreased culm number 1* (*dc1*) showed increased apical dominance of the main stem, and the growth of tillers was inhibited by the main stem [12]. Removal of the top of the main stem of the *dc1* mutant or knockout of the auxin-related gene *ABERRANT SPIKELET AND PANICLE 1* (*ASP1*) during the tillering stage can prevent the inhibitory effect of the main stem on tiller buds, thereby promoting their growth [12]. The critical genes involved in rice tiller differentiation include *DWARF3* (*D3*), *D10*, *D14*, *D27*, and *D53*. The proteins encoded by these genes carry out their functions through strigolactones (SLs), and mutants with function deletion of these genes exhibit stunted plant height and increased tiller numbers [13–17].

Tiller angle is another important agronomic trait of rice plant architecture [5]. The *PROSTRATE GROWTH 1* (*PROG1*) gene is a critical gene that controls the growth of wild rice (*Oryza rufipogon* L.) from prostrate growth to erect growth; this gene encodes a zinc finger transcription factor composed of 167 amino acids, and it is mainly expressed in axillary shoot meristems [18,19]. The loss function of *PROG1* not only improves plant structure but also increases the number of grains per panicles, thereby leading to a significant increase in the yield (manifested as "pleiotropism") [18,19]. *TILLER ANGLE CONTROLLING 1* (*TAC1*) is a major gene that regulates the tiller angle in rice. In one specific study, *TAC1* overexpression increased the tiller angle in rice, and plants with *TAC1*-RNAi showed a compact structure [20]. The genes *TAC1*, *TAC3*, and *D2* play important roles in the regulation of the tiller angle in rice; however, the molecular mechanisms underlying their roles in plant architecture remain unclear [21]. *TAC4* controls the tiller angle in rice by regulating indoleacetic acid content and affecting auxin distribution, thereby regulating the gravitational response of rice [22]. The *LAZY1* (*LA1*) gene is a negative regulator of auxin polar transport and regulates the tiller angle in rice [23–26]. *LA1* encodes a Cys(2)-His(2) zinc finger protein that functions as a transcription factor and binds to the *cis*-acting elements of its downstream genes to regulate their expression, thereby controlling tillers [23]. PROG1 acts upstream of the *LA1* gene as a repressor protein of rice and controls the tiller angle by regulating the geotropism of seedlings and the asymmetric distribution of auxin mediated by LA1 [27]. *LA1* gene overexpression partially rescues the defects of a larger tiller angle and stem orientation in *PROG1* complementary transgenic plants [27].

To summarize, many related studies on the tiller angle and tiller number in rice have been conducted in recent years. The identification of genes related to controlling these traits has facilitated the development of the ideal plant type and molecular design breeding in rice. However, the genetic regulatory network of many genes and their molecular mechanisms for regulating the tiller angle have not yet been fully understood, and further studies on this topic are required from different perspectives. We obtained a *multi-tillering and lazy 1* (*mtl1*) mutant from a *japonica* rice cultivar Wuyunjing 7 (WYJ7) via mutagenesis treatment with a heavy ion beam. In the present study, the *mtl1* mutant was used as the research material to conduct agronomic trait investigations, genetic law analysis, and fine mapping to (1) identify the regulation of rice tiller number and the tiller angle gene *MTL1*,

(2) explore the genetic network of plant architecture development, and (3) strengthen the theory of ideal plant type breeding in rice.

## 2. Materials and Methods

### 2.1. Material Planting and Investigation of Agronomic Traits

An *mtl1* mutant was obtained from the *japonica* cultivar WYJ7 via mutagenesis treatment with a heavy ion beam in 2015. The methods used for mutagenesis have been described by Xu et al. (2012) [28]. The parameters used in the present study were as follows: an ion energy of 25 keV; N$^+$ ion fluence of $2.5 \times 10^{16}$ ions/cm$^2$; an irradiation time of 10 min [28]. In May 2021, the seeds of WYJ7 and *mtl1* were germinated under dark conditions in an oven at 37 °C. After sprouting, the seeds were spread on the seedling bed at the Anhui Science and Technology University, Fengyang Campus, Anhui Province (32°52′30″ N, 117°33′15″ E). Once the seedlings reached the growth stage of four leaves and one heart, they were transplanted into rice fields consisting of rice plants with equal fertility and similar textures. The phenotypes of WYJ7 and *mtl1* were observed throughout the growth period. During the tillering stage, 15 plants of WYJ7 and *mtl1* with consistent growth stature were selected to determine their tiller angle and tiller number. During the maturity stage of rice, the plant height, panicle length, number of primary and secondary branches, and number of grains per panicle of WYJ7 and *mtl1* were counted. Grain length, grain width, and grain thickness were measured using a vernier caliper, and the 1000-grain weight and yield per plant were quantified. The obtained data were statistically analyzed via Student's *t*-test using SPSS software (Version 21.0, IBM SPSS), and graphs were created using GraphPad Prism software (FreeImage Public License—Version 1.0).

### 2.2. Construction and Genetic Analysis of the F$_2$ Segregated Population

The *mtl1* mutant was crossed with the *indica* rice cultivar Huajingxian 74 (HJX74) to obtain F$_1$ generation seeds, and the F$_1$ generation plants were self-crossed to obtain the F$_2$ generation segregated population. During the maturity stage, the number of individual plants with wild-type and multi-tillering and lazy phenotypes in the F$_2$ generation segregated population was determined, and the data were analyzed via the chi-square test using SPSS software (Version 21.0, IBM SPSS).

### 2.3. Mixed-Pool Sequencing and Single-Nucleotide Polymorphism Analysis

Twenty-five young leaves of each individual were collected from the HJX74 phenotypes and the extreme multi-tillering and lazy phenotypes (large tiller angle and unable to grow upright, like *lazy 1* mutant [23]) in the F$_2$ population of HJX74 and *mtl1*. The samples were sent to Wuhan Shuangluyuan Chuangxin Science and Technology Research Institute Co., Ltd. for bulked segregation analysis (BSA). First, two DNA mixing pools were constructed from the individuals of *mtl1* and wild-type phenotypes in the segregated population. By using the Japanese version 7.0 genome as a reference genome, the mixing pools were scanned using Green Super Rice 40K high-density rice gene chips containing 44,263 single-nucleotide polymorphism (SNP) checkpoints in 4726 cultivated rice varieties, and the allelic differences between the two mixed pools and their parents were analyzed. The gene segments related to the phenotypes were mapped to preliminarily locate the *MTL1* gene.

### 2.4. Fine Mapping of the MTL1 Gene

To perform fine mapping of the *MTL1* gene, we collected young leaves of each individual that exhibited *mtl1* traits from the F$_2$ segregated population hybridized between *mtl1* and HJX74, and a modified cetyltrimethylammonium bromide method was used to extract the DNA from the leaves [29].

Based on the nucleotide differences between Nipponbare and 93-11 reference genomes, insertion and deletion (InDel) primers (Table S1) were designed using the Primer 3 on-

line website (https://primer3.ut.ee/, accessed on 10 December 2021) and synthesized by GENERAL Biosystems Co., Ltd. (Chuzhou, China).

PCR amplification was performed on LongGene (T20, LongGene Scientific Instrument Co., Ltd., Hangzhou, China) in a 20 μL reaction volume containing 2 μL genomic DNA template (25 ng/μL), 10 μL 2× Santaq PCR mix (Vazyme Biotech Co., Ltd., Nanjing, China) and 200 nmol/L of each primer. The reaction procedure was as follows: denaturation at 95 °C for 3 min, followed by 35 cycles at 95 °C for 15 s, 55 °C for 15 s, and 72 °C for 5 s, and a final extension for 10 min at 72 °C. The PCR amplification products were analyzed via electrophoresis with 4% agarose gel.

### 2.5. Analysis of the Candidate Genes

Based on our fine mapping results, the genes in the localization interval were annotated using the Rice Genome Annotation Project Database website (http://rice.plantbiology.msu.edu/, accessed on 10 October 2022), and the gene functions in the target interval were preliminarily determined. Gene-specific primers (Table S1) were designed according to the reference genes in the target region annotated on the website. The expression period of the candidate genes was predicted based on information from the Rice Annotation Project Database website (https://rapdb.dna.affrc.go.jp/, accessed on 10 March 2023). RNA was extracted and reverse-transcribed, and semi-quantitative RT-PCR primers were used to analyze the difference in the expression levels between candidate genes in *mtl1* and WYJ7.

To determine the expression profiles of the candidate genes, RNA of WYJ7 and *mtl1* seedlings was extracted using TRIzol (Vazyme Biotech Co., Ltd., Nanjing, China), and cDNA was synthesized using MonScript™ RTIII All-in-One Mix with dsDNase (Monad Biotech Co., Ltd., Suzhou, China) in accordance with the manufacturer's instructions.

Semi-quantitative RT-PCR was performed using the gene-specific primers (Table S1). The PCR amplification profile was as follows: 94 °C for 2 min, followed by 22–28 cycles of 95 °C for 15 s, 55 °C for 15 s, 72 °C for 5 s, and a final extension at 72 °C for 5 min. The rice *OsActin* gene was used as the reference gene for the internal control.

## 3. Results

### 3.1. Phenotypic Analysis of the mtl1 Mutant

#### 3.1.1. Plant Architecture Analysis of *mtl1*

Under field cultivation conditions, the tiller angle of *mtl1* significantly increased during the vegetative growth period of rice (Figure 1a,b; $P = 2.06 \times 10^{-19}$), with an average tiller angle of approximately 110°, whereas the tiller of WYJ7 exhibited near-erect growth (Figure 1c). With the transition from the vegetative growth to reproductive growth stage, WYJ7 gradually stopped tillering, while *mtl1* continued to produce a high level of tillers. Before heading, the number of tillers in *mtl1* was significantly higher than that in WYJ7 (Figure 1c,d; $P = 4.02 \times 10^{-10}$). At the mature stage of rice, the plant height in *mtl1* was significantly lower than that in WYJ7 (Figure 1e,f; $P = 2.64 \times 10^{-8}$).

#### 3.1.2. Panicle Type Analysis of *mtl1*

The investigation of panicle morphology (Figure 2a) and yield (Figure 2b) indicated that, compared to WYJ7, the average panicle length of *mtl1* increased by 1 cm (Figure 2c). Moreover, the number of primary and secondary branches per panicle in *mtl1* significantly increased (Figure 2d,e; $P = 1.72 \times 10^{-8}$ and $1.80 \times 10^{-11}$, respectively), The average number of grains per panicle in *mtl1* also significantly increased ($P = 2.07 \times 10^{-8}$), with WYJ7 and *mtl1* having an average of 161 and 214 grains per panicle, respectively (Figure 2f). The evaluation of yield per plant showed that *mtl1* had a significantly higher yield than WYJ7 ($P = 1.27 \times 10^{-4}$); the average yield per plant of WYJ7 and *mtl1* was approximately 54 g and 100 g, respectively (Figure 2g).

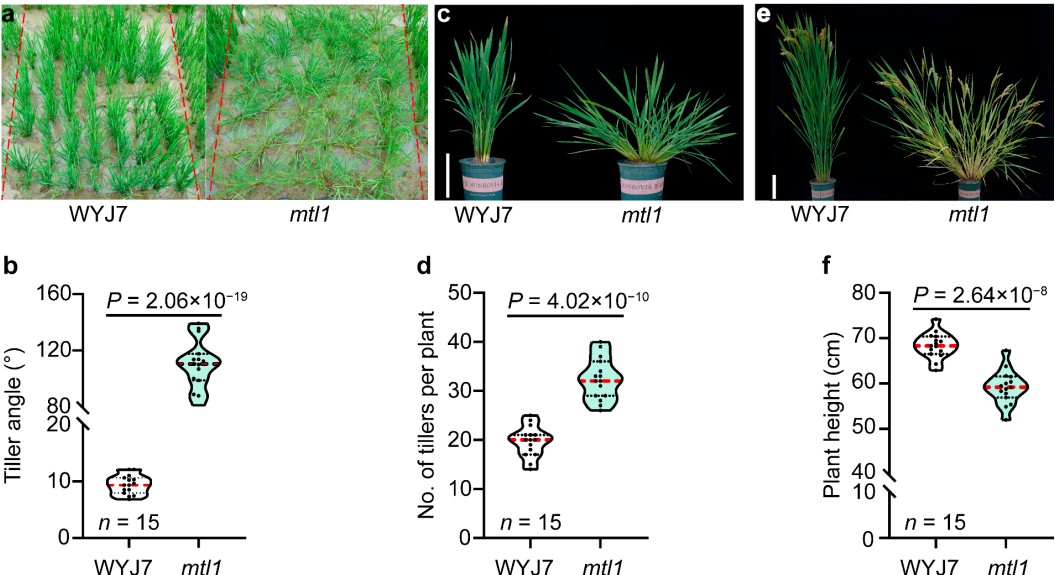

**Figure 1.** Comparison of plant types between WYJ7 and *mtl1* rice under field growth conditions. (**a**) Plant phenotypes grown under field conditions; (**b**) tiller angle; (**c,d**) tiller number; (**e,f**) plant height. Scale bar: 15 cm.

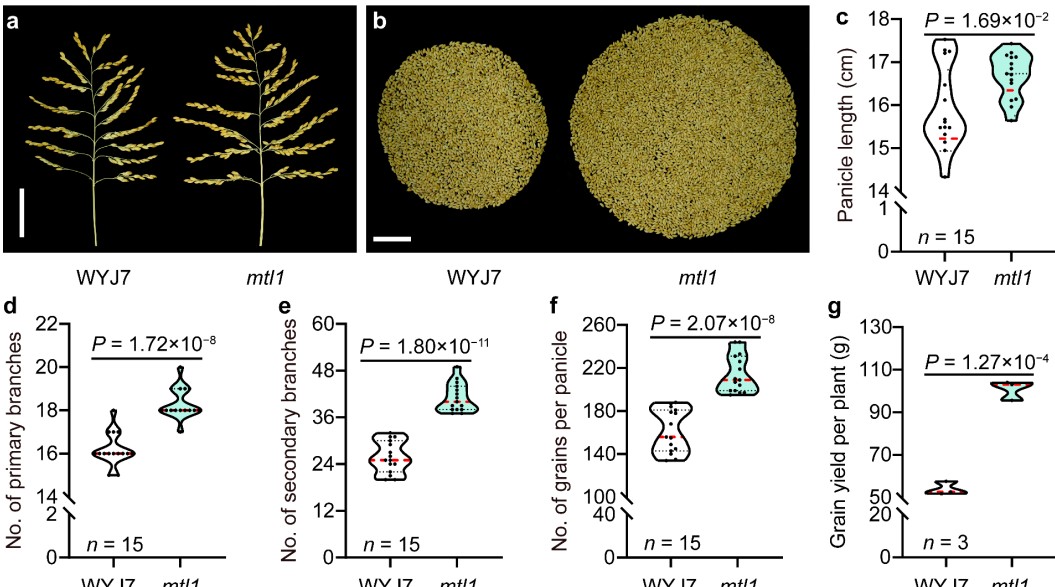

**Figure 2.** Comparison of panicle patterns between WYJ7 and *mtl1* rice. (**a**) Panicle type; (**b**) grains of an individual plant; (**c**) panicle length; (**d**) number of primary branches; (**e**) number of secondary branches; (**f**) number of grains per panicle; (**g**) grain yield per plant. Scale bar: 5 cm.

### 3.1.3. Grain Type Analysis of *mtl1*

We further compared the grain phenotypes, including grain length (Figure 3a), grain width (Figure 3b), and grain thickness (Figure 3c), between WYJ7 and *mtl1*. Compared to WYJ7, *mtl1* showed a moderate decrease in grain length ($P = 1.13 \times 10^{-2}$), which was approximately 98.79% of that of WYJ7 (Figure 3d). Grain width did not show significant differences between the two genotypes (Figure 3e; $P = 8.66 \times 10^{-1}$). However, the grain thickness of *mtl1* was significantly reduced ($P = 6.42 \times 10^{-3}$), which was approximately 98.04% of that of WYJ7 (Figure 3f). Additionally, the 1000-grain weight of *mtl1* was only 85.09% of that of WYJ7 (Figure 3g).

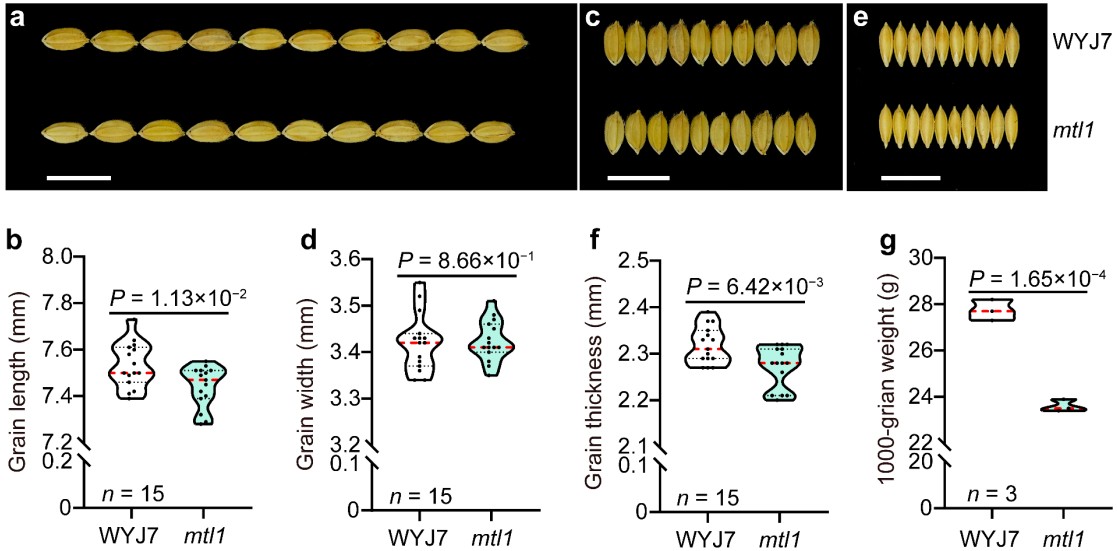

**Figure 3.** Grain type comparison between WYJ7 and *mtl1* rice. (**a**,**b**) Grain length; (**c**,**d**) grain width; (**e**,**f**) grain thickness; (**g**) 1000-grain weight. Scale bar: 1 cm.

### 3.2. Genetic Analysis of the mtl1 Mutant Phenotypes in Rice

To determine the genetic characteristics of the multi-tillering and lazy traits, we performed a genetic analysis of the *mtl1* mutant phenotypes. The observations revealed that the *mtl1* mutant phenotypes may be controlled by a recessive gene, as the $F_1$ generation plants resulting from crossing *mtl1* with HJX74 exhibited wild-type phenotypes. To clarify the genetic law of the *MTL1* gene, we performed a statistical analysis on the $F_2$ segregated population. The results showed a 3:1 segregation model between the wild-type and mutant phenotypes ($\chi^2 < \chi^2_{0.05, 1} = 3.84$) (Table 1), thus suggesting that the *mtl1* mutant phenotypes was controlled by a recessive gene.

**Table 1.** Genetic analysis of the *mtl1* mutant rice.

| Hybrid Crosses. | $F_1$ Phenotypes | $F_2$ | | | $\chi^2_{3:1}$ |
|---|---|---|---|---|---|
| | | HJX 74 Phenotypes | *mtl1* Phenotypes | Total Number of Plants | |
| *mtl1* × HJX74 | HJX74 | 505 | 161 | 666 | 0.20 |

### 3.3. Preliminary Mapping of the MTL1 Gene

To mine the *MTL1* gene that controls the multi-tillering and lazy phenotypes, we used gene chips to preliminarily map the *MTL1* gene by using the BSA method. According to the SNP chip scanning results, the genotype of the parent $P_1$--*mtl1* was almost of the same color (gray); the genotype of $P_2$-HJX74 was also almost of the same color (red), thus indicating that the parents had homozygous genotypes (Figure 4a). Moreover, in the $F_2$ generation segregated population, most of the genotypes in the WT mixed pool $F_2$-HJX74 and the *mtl1* mutant phenotypes mixed pool $F_2$-*mtl1* were almost heterozygous (blue); however, some segments on chromosome 11 showed differences between the two mixed pools.

A comparison of genomic differences between the two samples in the preliminary localization segment found that the genotype of the $F_2$-*mtl1* mixed pool was consistent with the *mtl1* mutant genotype (AA) (Figure 4b), while the genotype of the $F_2$-HJX74 mixed pool was predominantly heterozygous (AB) (Figure 4b,c). Based on the principle of BSA and the SNP loci of this differential segment, the results indicated that the gene related to the *MTL* phenotypes was preliminarily mapped between 5.58 and 17.64 Mb on chromosome 11.

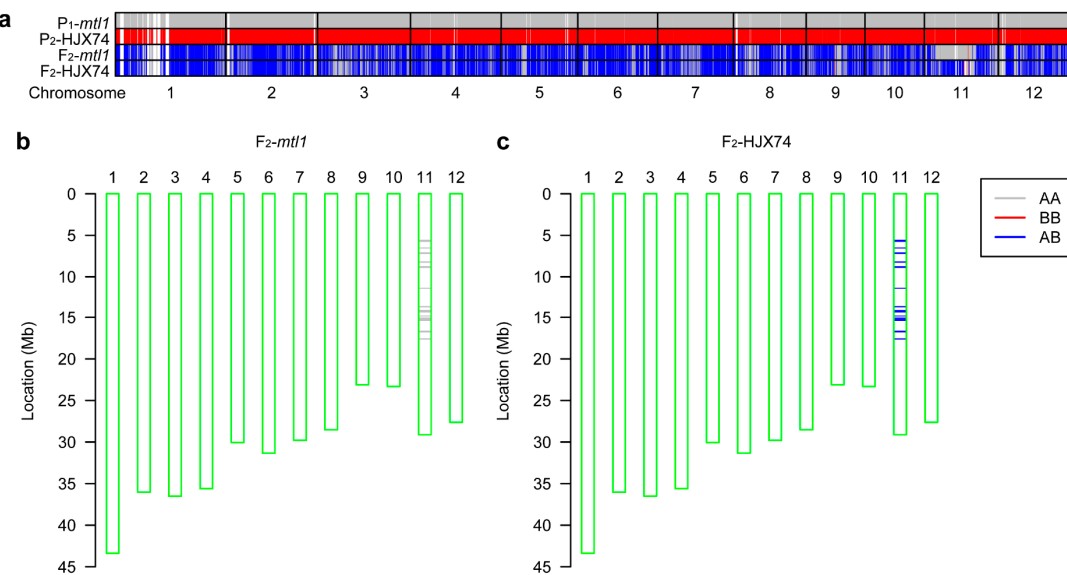

**Figure 4.** Preliminarily mapping of the *MTL1* gene using the BSA method. (**a**) Genotype analysis of parents and the mixed pools of F$_2$; (**b**,**c**) SNP analysis of the differential loci. Homozygous gene loci in the female parent P$_1$-*mtl1* are shown in gray (AA); homozygous loci in the male parent P$_2$-HJX74 are shown in red (BB); heterozygous gene loci in the segregated populations are shown in blue (AB).

### 3.4. Fine Mapping of the MTL1 Gene

According to the preliminary mapping, eight pairs of InDel primers (M1–M8) were first developed (Table S1), and the 230 individual plants of the *mtl1* mutant phenotypes from the F$_2$ population were used for *MTL1* gene mapping. Twenty-six crossing-over individual plants were found within the preliminary interval. Based on these 26 individual plants, the *MTL1* gene was then narrowed down to be between markers M6 and M8 (Figure 5a). Four polymorphic primers (M9–M12) were developed between M6 and M8, and six crossing-over individual plants between M6 and M8 were used to narrow down the *MTL1* gene between M12 and M8 (Figure 5b). According to the Rice Genome Annotation Project database website, the target interval was 66.67 kb and contained five putative genes (Figure 5c).

### 3.5. Analysis of the Candidate Genes

Based on the annotation results of the Rice Genome Annotation Project database, the target interval contains five genes (Table S2), among which three genes (*LOC_Os11g29810*, *LOC_Os11g29820*, and *LOC_Os11g29830*) encode predictive proteins of unknown functions; *LOC_Os11g29850* encodes an ATP-binding cassette transporter, and the *LA1* gene (*LOC_Os11g29840*) controls the rice tiller angle (Figure 5, Table S2).

To analyze the nucleotide differences of these five genes between WYJ7 and *mtl1*, we performed genome PCR assays using gene-specific primers (Table S1). The results showed that all five candidate genes were amplified normally in WYJ7. However, *LA1* could not be amplified normally in *mtl1*. Sequence comparisons involving these genes showed that *LOC_Os11g29840* was polymorphic in mapping parents and Nipponbare. Next, five exons of *LA1* were PCR amplified using WYJ7 and *mtl1* genomic DNA as templates. The five exons were easily amplified from WYJ7; however, no amplification products were detected for each exon from *mtl1*, thus suggesting that the *MTL1* gene was deleted in *mtl1* (Figure 5d). Semi-quantitative RT-PCR also confirmed that the *LA1* gene was not expressed in *mtl1* (Figure 5e).

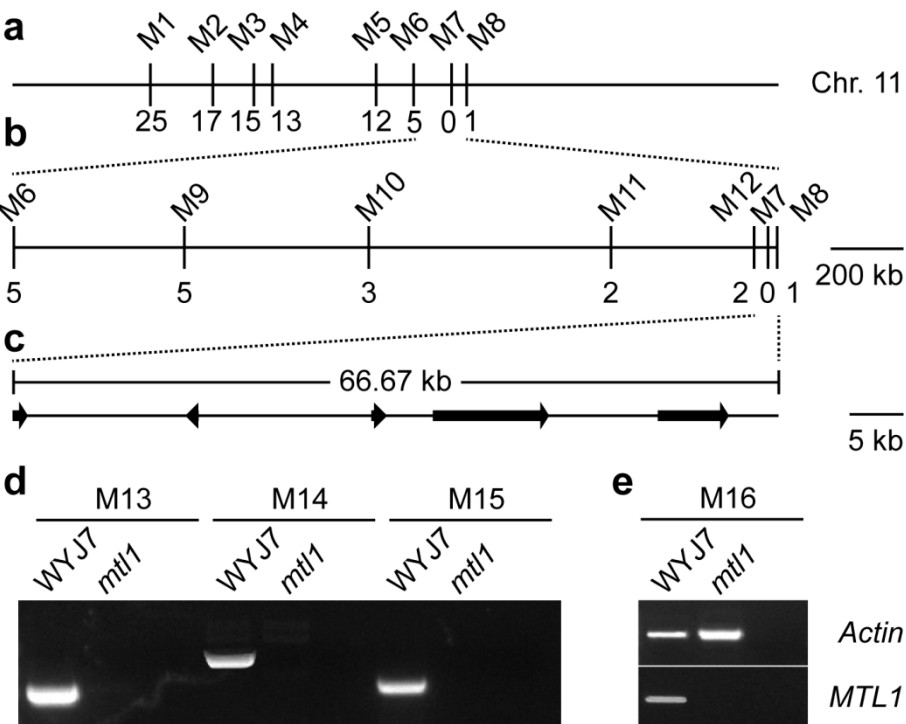

**Figure 5.** Fine mapping and candidate gene analysis. (**a**) Preliminary mapping of *MTL1*; (**b**) fine mapping of *MTL1* by using newly designed InDels; (**c**) candidate genes in the mapping interval. (**d**) PCR amplification of the exons of *LA1*; (**e**) semi-quantitative RT-PCR analysis of *LA1* expression level. Ms represent polymorphic markers; the numbers below the lines represent the number of recombinants. The arrow indicates the site of the predicted genes in the M8 to M12 interval.

## 4. Discussion

Tiller is one of the key factors for developing an ideal rice plant type. The tiller angle determines the planting density of rice and affects the yield by influencing photosynthetic efficiency [30,31]. During the domestication process of rice, the rice plant underwent a process from creeping growth to erect growth. The creeping growth pattern of wild rice is beneficial, allowing rice plants to compete with weeds for growth space in the wild environment, while the erect growth pattern of cultivated rice is beneficial for improving plant density and yield under agricultural conditions [32]. The results of domestication and artificial selection are mainly driven by C2H2-type zinc finger transcription factors, including PROG1, PROG7, and RICE PLANT ARCHITECTURE DOMESTICATION (RPAD) [18,19,33,34]. The change in the growth pattern of the Asian cultivated rice from creeping to erect is mainly caused by allelic mutations in the *PROG1* gene and a deletion of 110 kb in the *RPAD* gene [18,19,34], while the change in the African cultivated rice is mainly due to allelic mutations in the *PROG7* gene and a deletion of 113 kb in the *RPAD* gene [33,34]. The *TAC1* gene shows a low expression level in *japonica* rice, which exhibits a more compact plant architecture than *indica* rice and has been widely planted in high-latitude temperate regions and high-altitude regions [20,35]. Although many genes controlling rice tillers have been identified in recent years, only a few genes have been studied for their application in rice production. Therefore, there is an urgent requirement to explore new plant type control genes and enrich the theory of ideal plant types.

In the present study, the investigation of the agronomic traits of *mtl1* showed that while the tiller angle of *mtl1* increased significantly, the number of effective panicles did not continue to increase with the increase in tiller number (Figure 1a–e). The grain thickness and 1000-grain weight in *mtl1* decreased significantly (Figures 2 and 3). However, the panicle length, number of primary and secondary branches, and number of grains in *mtl1* increased significantly; therefore, yield per plant increased significantly. Although the yield

per plant of *mtl1* increased significantly, the large tiller angle presented a disadvantage in the form of an inability to increase plant density. Therefore, balancing the relationships between tiller angle, tiller number, and planting density and the relationship between grain number and grain size are urgent issues that need to be addressed in the design of an appropriate breeding approach and development of an ideal plant type.

　　We narrowed down the *MTL1* gene to the range of 17.27–17.33 Mb (66.67 kb) on chromosome 11 by using map-based cloning (Figure 5a–c). Our analysis of the candidate genes in the mapping interval revealed that the cloned gene *LA1* was missing (Figure 5d,e). Previous studies have shown that *LA1* regulates the polar transport of auxin and controls the gravitropism of the aboveground part of rice and ultimately the tiller angle. Exon 4 of the *lazy1-ZF802* mutant in rice has a deletion of 8 bp, leading to the premature termination of protein translation, and the entire gene of the *lazy1-Shiokari* mutant is missing; both these plants exhibit the same dispersed tillering phenotypes [23]. By carrying out the appropriate knockdown of *LA1* gene expression by using RNAi, the tiller angle can be moderately increased, and the RNAi plant phenotypes of the *LA1* gene are consistent with the *la1* mutant phenotypes [23]. Brevis Radix Like 4 (OsBRXL4) interacts with LA1 and affects its nuclear localization, thereby influencing the gravity response and tiller angle of rice. The overexpression of *OsBRXL4* can reduce LA1 localization in the nucleus, thereby increasing the tiller angle and leading to semi-scattered phenotypes in rice [36]. These results showed that the tiller angle of rice is associated with the expression level of the endogenous *LA1* gene. In rice, the loss of *LA1* function increased the tiller angle; however, it did not cause any significant changes in panicle morphology. In maize, *ZmLA1* gene mutation not only formed a creeping main stem but also affected the morphology of corn tassels and ears, as noted by the authors of [36]. In another study, the insertion of a transposon *MuDR* at 25 bp of the fourth exon of *ZmLA1* significantly decreased the expression level of *LA1*, thereby resulting in a reduced number of tassel branches, a decrease in pollen shedding and ear silk, irregular axis arrangement, and reduced seed setting rate [37]. In the present study, compared to the *la1* mutant, the deletion of the *MTL1* gene also produced a prostrate growth pattern. However, *mtl1* showed a larger panicle and an increase in grain yield per plant, which have not been reported for the existing *la1* mutant. Considering our results in tandem with the findings of previous studies, we speculate that the *mtl1* mutant phenotypes are caused by the deletion of the *LA1* gene. A recently published study revealed that in *Arabidopsis*, gravistimulation triggers and regulates the MAPK-mediated phosphorylation of LA proteins [38]. Phosphorylated LA proteins enhance their interaction with the TOC proteins on the surface of amyloplasts, leading to the enrichment of LA proteins on the surface of amyloplasts, asymmetrical auxin distribution, and root differential growth; the shuttling of LA proteins from amyloplasts to the adjacent plasma membrane triggers gravity sensing in plants [38]. So, how does the complete deletion of LA proteins in rice affect auxin polar transport? The *mtl1* mutant could be a potentially good genetic material to understand this aspect.

　　Thus far, the regulatory networks and signaling pathways associated with *LA1* controlling the tiller angle of rice are largely unknown. Moreover, the limited progress made in identifying new genes hinders their applications in the structural improvement and elucidation of the molecular mechanisms underlying tiller angle regulation.

　　To summarize, in the present study, we found that the different mutation types of *LA1* play a very important role in controlling the tiller angle of rice. In the future, CRISPR-Cas9 technology could be used to conduct in-depth studies on the different mutations of *LA1*, along with the aggregation of other genes, to create a suitable tiller angle and ideal plant type for rice plants.

**Supplementary Materials:** The following supporting information can be downloaded at: https://www.mdpi.com/article/10.3390/agriculture13112166/s1, Table S1: Primers used in the experiments; Table S2: Gene annotations in the fine mapping interval.

**Author Contributions:** X.H. and B.L. designed the work; Z.S. and T.M. performed experiments and wrote the original manuscript. T.F., H.A. and Y.Y. contributed to genetic hybridizations and the analysis of experimental data; S.D. helped with plant materials in the fields; X.H. and B.L. revised the manuscript. All authors have read and agreed to the published version of the manuscript.

**Funding:** This study was supported by the Excellent Scientific Research and Innovation team of the Education Department of Anhui Province (2022AH010087), the Anhui Province Key Research and Development Program (202204c06020070), the Talent Introduction Start-up Fund Project of Anhui Science and Technology University (NXYJ202001), and the Industry University Research Cooperation Project Shouxian Branch Center for the Transformation of Scientific and Technological Achievements of Anhui University of Science and Technology (kycg881215).

**Institutional Review Board Statement:** Not applicable.

**Informed Consent Statement:** Not applicable.

**Data Availability Statement:** The data reported in this study are available in the article and supplementary materials.

**Conflicts of Interest:** The authors declare no conflict of interest.

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
