# Peer review of "Deletion of the OsLA1 Gene Leads to Multi-Tillering and Lazy Phenotypes in Rice"

_agriculture, doi:10.3390/agriculture13112166_

Round 1

Reviewer 1 Report

Comments and Suggestions for Authors

Dear Editor,

I wish to thank you for the opportunity to review this manuscript. In my opinion, the current submission, "Deletion of the OsLA1 Gene Leads to Multi-Tillering and Lazy Phenotypes in Rice," looks simple, but it is interesting and generally well-written. In the discussion, the author mentioned that the "number of grains and yield per plant in mtl1 increased significantly; however, the grain length, grain width, grain thickness, and 1000-grain weight decreased significantly." How did the author measure the yield? There is a contradiction in the results, which should be clarified.

In the abstract part, what the author means by "MTL1 gene was narrowed!" and "rice plant types"

In the Materials and Methods part, the author mentioned that the rice mutant was created by a heavy ion beam in 2015; however, no details or references were mentioned. Why did the author use this cultivar, Huajingxian 74, for creating the F2 mapping population? Also, why was semiquantitative RT-PCR used instead of quantitative real-time PCR?

The results part is well described and presented; however, the wild-type phenotype means what? The phenotype of HJX74 or what? It is not clear!

The discussion part is also well-written; however, the author used "ear morphology" in rice. Is it correct to use the word ear?

Finally, there are some typing mistakes; for example, "the multi-tillering and lazy mutant phenotype was regulated", The word "phenotypes" should be used.

  Comments on the Quality of English Language

Some typing mistakes are present.

Reviewer 2 Report

Comments and Suggestions for Authors

The present work “Deletion of the OsLA1 Gene Leads to Multi-Tillering and Lazy 2 Phenotypes in Rice 3” by  Sun et al, Showed the isolation of a multi-tillering and lazy mutant from the japonica rice cultivar WYJ7, achieved through treatment with a heavy ion beam. Both phenotypic and genetic analyses revealed that the multi-tillering and lazy mutant phenotype is governed by a recessive gene referred to as Multi-Tillering and Lazy 1 (MTL1), which is allelic to the LAZY1 (LA1) gene. Molecular analysis conducted on the segregating population further confirmed its location on chromosome 11, within a narrow range of 66.67 kb. The manuscript is well-crafted and presents the findings effectively. I recommend its publication in its current form

Comments on the Quality of English Language

Minor editing of English language required

Reviewer 3 Report

Comments and Suggestions for Authors

The manuscript “Deletion of the OsLA1 Gene Leads to Multi-Tillering and Lazy Phenotypes in Rice” is a mutation-based fine-mapping study of the tiller angle controlling gene. It is a good study and can be accepted after minor revision.

Improve the grammar of the manuscript. There are many grammar mistakes, extra commas, and wrong sentence formation.

1.       Rephrase the first sentence of the abstract, and remove the “etc”.

2.       The process of mutation and selection is missing. Add the method of how the mutant was developed and evaluated. How many seeds were treated? What was beam intensity? What was the germination rate from treated and non-treated? Explain all this.

3.       Line 127, please tell something about LAZY phenotypes. All of a sudden where it comes from?

4.       Line 142, what is Nipponbare?

5.       Line 247, add supplementary data about these results

6.       Line 263-265, add data about these results as supplementary figures.

Comments on the Quality of English Language

The English in the abstract should be improved. The English language of the manuscript is OK. 
